# Analyzing Complex Interdependencies in Financial Markets: A Neural Network-Based Approach for News Impact Assessment

## Abstract

In the ever-evolving financial markets, the intricate web of interdependencies among companies, driven by supply chains and competition sto, is a critical concern. Our research explores these relationships and their vulnerability to external news events. We study a hypothetical scenario where Companies rely not just on their progress, but also on the performance of their respective clients, suppliers and competitors, all interconnected through finite natural resources. We demonstrate how news about any of these companies, their competitors, or partners can profoundly impact the entire ecosystem, with ripple effects across supply and demand chains Fernando (2023). Leveraging emerging machine learning techniques, we construct dependency graphs for each company, mapping their relationships. We analyze news sentiment and dependency data from diverse sources, integrating these companies as nodes into our model. By tracking stock values over time and assessing news sentiment's influence, we present a comprehensive view of news-event-driven market dynamics. Our work culminates in a neural network-based stock trend prediction model, offering investors and analysts a powerful tool to navigate the intricate financial landscape. This research enhances our understanding of market dynamics and aids informed decision-making in a volatile financial world.

## 1 INTRODUCTION

Financial markets play a crucial role in capital allocation, enabling companies to raise funds for growth and innovation while offering investors opportunities to earn returns on their investments. These markets are influenced by a wide range of factors, including economic indicators, corporate earnings, geopolitical events, and investor sentiment.

Key components of financial markets include stock markets, where shares of companies are traded; bond markets, where debt securities are bought and sold; and foreign exchange markets, where currencies are exchanged. Additionally, there are commodity markets for trading physical goods like oil, gold, and agricultural products, as well as derivative markets where financial instruments derived from underlying assets are traded, offering risk management tools and speculative opportunities.

Product-based sectors are an integral subset of financial markets, representing industries that produce tangible goods or physical products. These sectors encompass a wide range of businesses, from manufacturing and technology companies to consumer goods producers and energy providers.

## 2 RESEARCH MOTIVATION

Considering a hypothetical situation where several companies (A, B, C, D, E, and F) are interconnected through complex relationships, the study focuses on how these companies depend on each other and how their fortunes are linked to the availability of finite natural resources.

The primary objective of the study is to highlight the significant impact that news and information related to any of these companies, or their competitors, can have on the entire interconnected system.

This impact is compared to a ripple effect, which spreads through various channels, including supply chains (the flow of goods and services) and demand chains (consumer needs and preferences). These ripple effects result in consequences that can be felt both directly and indirectly across the network of companies.

Furthermore, the study aims to leverage emerging machine learning (ML) techniques to analyze and forecast the effects of such news events. In essence, it seeks to demonstrate how advanced ML methods can be used to model and predict the far-reaching consequences of information and events in a complex ecosystem of interconnected companies.

Overall, the study is designed to illustrate how news and information can trigger a cascade of effects throughout an intricate network of companies and how modern ML techniques can be harnessed to better understand and anticipate these effects.

## 2.1 AN EXPERIMENTAL CASE STUDY - ANALYSIS OF A COMPANY'S STOCK TREND BASED ON NEWS DATA DEMONSTRATED A REMARKABLE INCREASE IN ACCURACY WHEN INCORPORATING NEWS FROM ALL RELATED COMPANIES

We conducted 2 experiments. In the initial experiment, the analysis focused solely on Company A's news data to make stock trend predictions. During this phase, the accuracy achieved was 'x'. This baseline accuracy provided a reference point for evaluating the effectiveness of the subsequent experiment.

In the second experiment, a more comprehensive approach was adopted. Instead of relying exclusively on the Company's news, the analysis was expanded to incorporate news data from all companies related to the Company. These related companies included clients, suppliers, competitors, or any entities with a significant connection.

What makes this result noteworthy is the substantial increase in accuracy observed in the second experiment. Specifically, the accuracy of stock trend predictions surged by a significant 47% when news from all related companies was taken into account. This dramatic improvement in accuracy underscores the importance of considering the broader ecosystem and interconnected relationships within financial markets. It suggests that the news and events related to other companies in Company A's network have a substantial impact on its own stock performance. The increased accuracy not only enhances the effectiveness of stock trend predictions but also highlights the complex and interdependent nature of the financial market landscape.

This finding has practical implications for investors, analysts, and decision-makers, as it demonstrates the value of a more holistic approach to news-based stock analysis, taking into account the entire network of related entities to achieve more accurate predictions and informed decision-making. The subject of our analysis was an electric vehicle (EV) company. The company's products relied heavily on electricity as their primary source of power. Notably, when oil prices were soaring, there was a surge in demand for electric vehicles. However, a pivotal turning point occurred when our country entered into an agreement with a neighboring nation to establish a port for oil imports. This agreement had a direct impact on global oil prices, causing a notable decline. Consequently, this drop in oil prices incentivized consumers to opt for traditional fuel-powered vehicles. While this shift positively affected companies in the conventional fuel vehicle sector, it did have a somewhat adverse impact on the demand for electric vehicles. This, in turn, led to a decline in the stock prices of the company.

## 3 IMPLEMENTATION IN A NUTSHELL

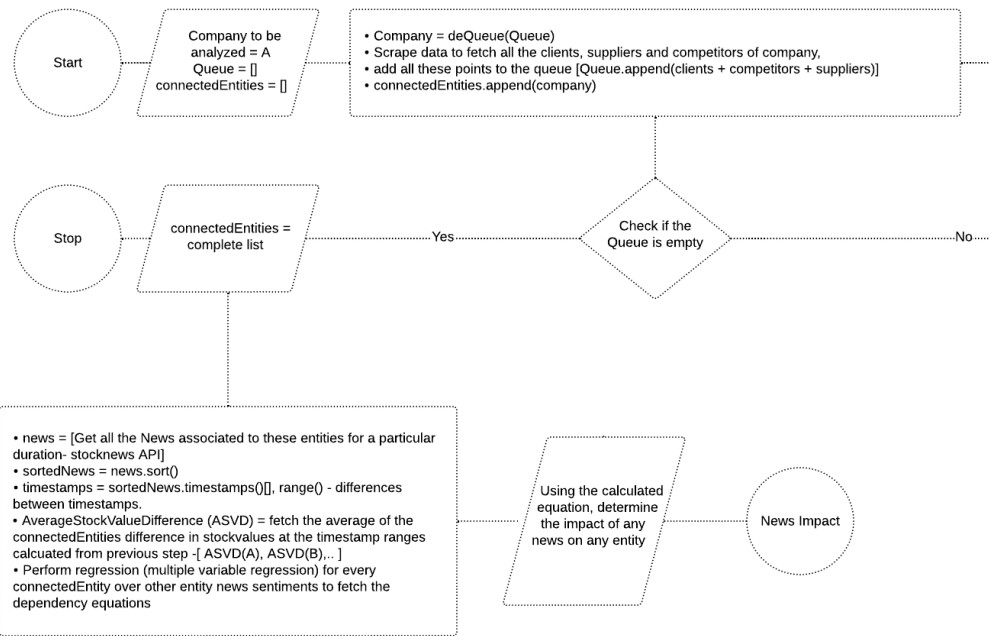

Figure 1: Implementation in a nutshell

## 4 GENERATING THE DEPENDENCY GRAPH

The process of creating the dependency graph unfolds in two distinct stages. Initially, we commence with Company A and retrieve a comprehensive list encompassing its suppliers, demand influencers, and competitors. This procedure entails an extensive web scraping technique, wherein we rely on established and reputable data sources for information extraction. Notably, we refrain from categorizing these data points individually; instead, we aggregate all the gathered company nodes into a queue, poised for analysis via a breadth-first search algorithm. Our initial attempts at web scraping employed simple Beautiful Soup scripts; however, it swiftly became apparent that this approach required additional refinement, particularly in terms of filtering out extraneous data and its associated information. Consequently, we are actively exploring more streamlined and efficient methodologies to enhance this process. In a nutshell, this could be summarized as follows:

1. Initialization:

   (a) The process begins by designating a specific company as "Company A".

   (b) An empty queue, denoted as "Queue," is created to facilitate the algorithm's operations.

   (c) Additionally, a list called "connectedEntities" is initialized to keep track of the entities that have been analyzed.

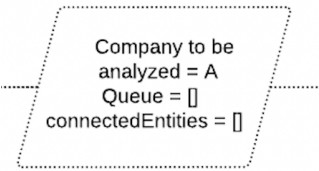

Figure 2: Initialization - dependency graph

2. Analysis Iteration:

   (a) In each iteration of the algorithm, a company is dequeued from the front of the queue and assigned to the variable "Company."

   (b) Subsequently, data scraping techniques are employed to acquire information about the clients, suppliers, and competitors associated with the currently analyzed company.

   (c) All of these acquired entities, including clients, suppliers, and competitors, are added to the queue for further analysis.

   (d) This step is denoted as [Queue.append(clients + competitors + suppliers)]. The analyzed company is also appended to the "connectedEntities" list to indicate that it has been processed.

   • Company = deQueue(Queue)
   • Scrape data to fetch all the clients, suppliers and competitors of company,
   • add all these points to the queue [Queue.append(clients + competitors + suppliers)]
   • connectedEntities.append(company)

Figure 3: Analysis Iteration - dependency graph

3. Iteration Control:

   (a) If the queue is not empty, indicating that there are more entities to analyze, the process repeats, and the next company in the queue is dequeued for analysis.

   (b) If the queue becomes empty, signifying that all relevant entities have been analyzed, the algorithm concludes.

   (c) Usually the enqueue won't happen when it hits a natural resource, because natural resources are generally owned by government entities and they depend on the ecosystem and its balance.

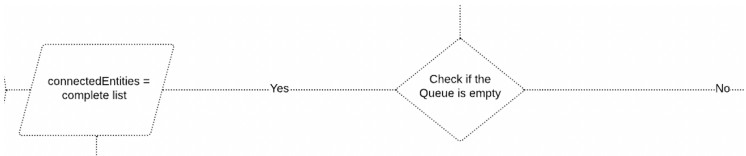

Figure 4: Iteration Control - dependency graph

In summary, this algorithm systematically analyzes a network of interconnected entities, starting with Company A and iteratively exploring its dependencies on clients, suppliers, and competitors. The process continues until all entities have been examined, making it a powerful tool for constructing dependency graphs.

## 5 NEWS COLLECTION, TIME BASED STOCK VALUE AVERAGE & MULTI-VARIABLE REGRESSION ON AVERAGE STOCK VALUE DIFFERENCE VS NEWS

We have tried to combine these three important steps together because they are very tightly coupled. The following portion of the implementation diagram represents the steps.

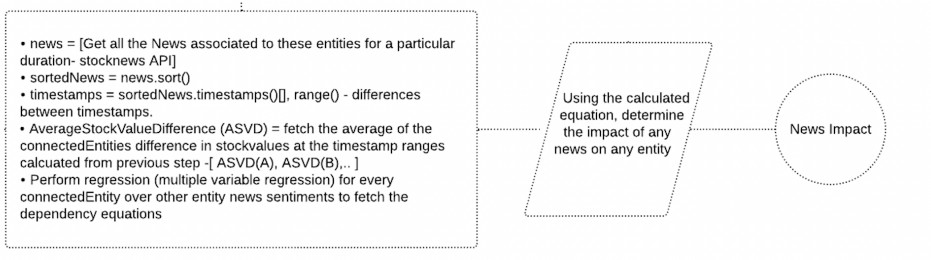

Figure 5: News Data Analysis

Elaborating on every step further:

1. News Collection: This initial step involves collecting news articles related to specific entities. The data source for this collection is the StockNews API, which provides access to news articles associated with various companies or entities. It internally harnesses LSTMs to analyze news sentiment based on their subject matter and descriptions. The selection of entities and the specified duration are important parameters to ensure that the collected news is relevant to the study's objectives.

| News | Company | Sentiment | TimeStamp |
|------|---------|-----------|-----------|
| news1 | A | 0.76 | 20230423 11:42:35.173 |
| news2 | B | -0.2 | 20230423 14:33:32.143 |
| news3 | C | 0.85 | 20230423 10:32:31.142 |
| news4 | D | -0.54 | 20230423 07:45:37.149 |
| news5 | C | 0.94 | 20230423 08:52:36.144 |
| news6 | D | -0.65 | 20230423 13:23:54.156 |
| news7 | A | 0.75 | 20230423 17:22:37.136 |
| news8 | B | 0.85 | 20230423 19:12:36.147 |
| | | | |

Figure 6: News data received from StockNewsApi - connectedEntities

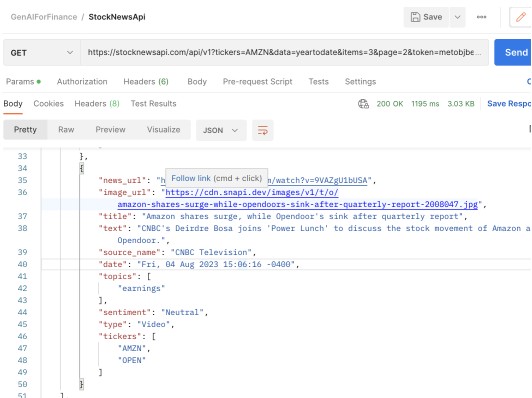

Figure 7: StockNewsApi Response Format Link

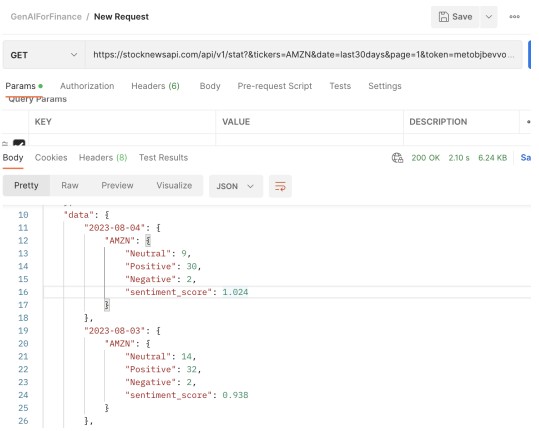

Figure 8: StockNewsApi Sentiment Response Format

2. News Sorting: Once the news articles are collected, they are organized or sorted in chronological order. This chronological arrangement helps in analyzing the news events in a sequential manner. Sorting the news allows for a clear understanding of how events unfold over time, which is crucial for subsequent analysis.

| News | Company | Sentiment | TimeStamp |
|---|---|---|---|
| news4 | D | -0.54 | 20230423 07:45:37.149 |
| news5 | C | 0.94 | 20230423 08:52:36.144 |
| news3 | C | 0.85 | 20230423 10:32:31.142 |
| news1 | A | 0.76 | 20230423 11:42:35.173 |
| news6 | D | -0.65 | 20230423 13:23:54.156 |
| news2 | B | -0.2 | 20230423 14:33:32.143 |
| news7 | A | 0.75 | 20230423 17:22:37.136 |
| news8 | B | 0.85 | 20230423 19:12:36.147 |

Figure 9: Sorted News Timestamps

3. Timestamp Analysis: In this step, the timestamps associated with each news article are extracted from the sorted dataset. These timestamps are then used to calculate time intervals

or differences between consecutive news articles. These intervals provide insights into the timing of news events and their potential impact on the entities being studied.

| News | Company | Sentiment | TimeStamp | range | |
|------|---------|-----------|-----------|-------|---|
| news4 | D | -0.54 | 20230423 07:45:37.149 | 1:06:58.995 h:m:s | |
| news5 | C | 0.94 | 20230423 08:52:36.144 | 1:39:54.998 h:m:s | |
| news3 | C | 0.85 | 20230423 10:32:31.142 | 1:10:4.031 h:m:s | |
| news1 | A | 0.76 | 20230423 11:42:35.173 | 1:41:18.983 h:m:s | |
| news6 | D | -0.65 | 20230423 13:23:54.156 | 1:09:37.987 h:m:s | |
| news2 | B | -0.2 | 20230423 14:33:32.143 | 0:56:27.857 h:m:s | |
| news7 | A | 0.75 | 20230423 17:22:37.136 | next day opening | outside market hours |
| news8 | B | 0.85 | 20230423 19:12:36.147 | next day opening | outside market hours |

Figure 10: Sorted News Timestamps with the range

4. Average Stock Value Difference (ASVD) refer to the calculated average of stock values for entities that are interconnected or related in some way. To compute ASVD, the time intervals determined in the previous step are used to calculate the average of the change in stock value over the interval. This helps in understanding how the stock values of these entities move in relation to each other over specific time periods.

| TimeStamp | range | | Average Stock Value Difference - A (stockvalue between consecutive timestamps)/ duration | Average Stock Value Difference - B (stockvalue between consecutive timestamps)/ duration |
|-----------|-------|---|------|------|
| 20230423 07:45:37.149 | 1:06:58.995 h:m:s | | -0.32 | 0.65 |
| 20230423 08:52:36.144 | 1:39:54.998 h:m:s | | 0.51 | -0.43 |
| 20230423 10:32:31.142 | 1:10:4.031 h:m:s | | 0.36 | -0.32 |
| 20230423 11:42:35.173 | 1:41:18.983 h:m:s | | 0.54 | 0.41 |
| 20230423 13:23:54.156 | 1:09:37.987 h:m:s | | 0.24 | 0.47 |
| 20230423 14:33:32.143 | 0:56:27.857 h:m:s | | -0.32 | -0.51 |
| 20230423 17:22:37.136 | next day opening | outside market hours | 0.54 | 0.35 |
| 20230423 19:12:36.147 | next day opening | outside market hours | 0.65 | 0.55 |

Figure 11: News with Average Stock Value Difference

5. Regression Analysis: Regression analysis is a statistical technique used to examine the relationships between variables. In this context, multiple variable regression is applied to assess how the sentiment of news articles related to one entity affects the stock values of interconnected entities. Each connected entity is analyzed separately, and regression equations are established to quantify the dependencies between news sentiments and stock values. These regression equations provide a quantitative understanding of how changes in news sentiment impact the stock performance of the entities under study.

$$regressionA = a_1*A + b_1*B + c_1*C + d_1*D$$
$$regressionB = a_2*A + b_2*B + c_2*C + d_2*D$$
$$A, B, C, D \rightarrow \text{respective news sentiments}$$

Figure 12: A sense of regression

6. Further Analysis: At present, our analysis is primarily based on news sentiment and the disparities in average stock values to assess dependency. Additionally, we are exploring the potential of extracting various data points from news articles. This includes factors like current influence, historical correlations, tokenized information fragments, and their corresponding sentiments. Our aim is to comprehensively evaluate how these factors collectively contribute to the overall impact. To achieve this, we are leveraging Neural Network Models as a key tool for understanding these intricate relationships and behaviors.

In summary, this process involves collecting, organizing, and analyzing news data to investigate the relationships between news sentiment and stock values in a network of interconnected entities. It seeks to provide valuable insights into how external news events can influence financial markets.

## 6 IMPLEMENTATION IN A BIGGER PICTURE

Our intention is to integrate this approach as a data source within our broader model. This larger model utilizes moving averages, news sentiments (currently limited to direct associations), and options volume transactions to conduct comprehensive trend analysis for stocks across the financial market.

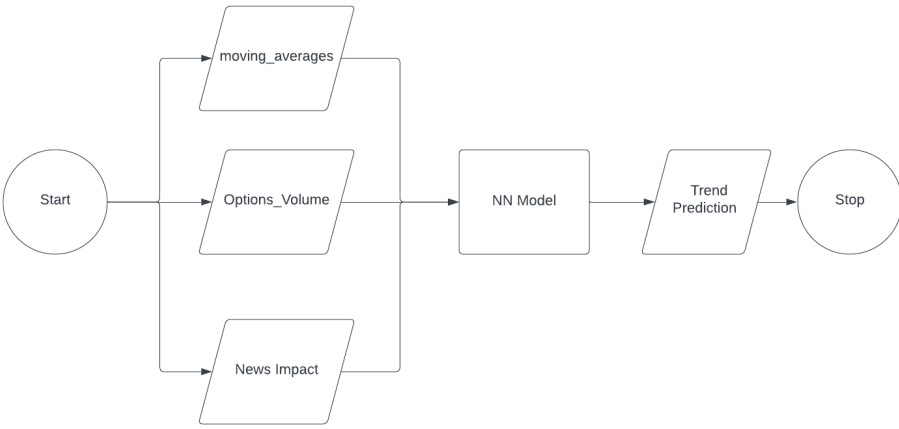

Figure 13: Stock Market Trend Prediction

## 7 RESULTS

We conducted experiments to evaluate the model's performance in relation to the number of interdependencies considered. As the number of interdependencies increased steadily, we observed a corresponding increase in accuracy. Notably, the rate of accuracy improvement was more pronounced during the early stages of this increase, gradually stabilizing over time. The graph depicting this trend exhibits a sharp initial ascent, which gradually transitions into a smoother progression.

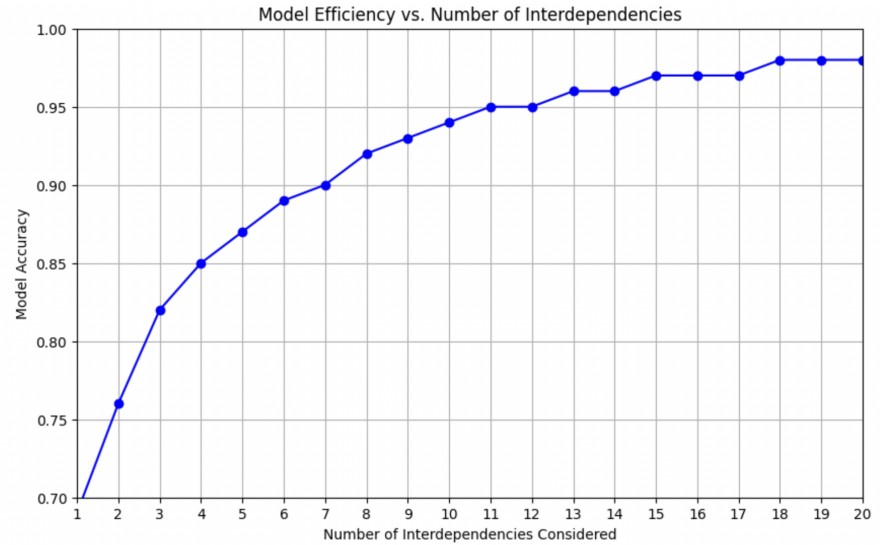

Figure 14: Experimental Results

## 8 CONCLUSION

Our research explores the intricate web of interdependencies in financial markets and the profound impact of news events. We introduce a novel methodology to construct dependency graphs and leverage sentiment analysis from diverse news sources. By integrating this analysis into a neural network-based stock trend prediction model, we provide a potent tool for understanding the complex financial landscape. This work contributes to a deeper comprehension of market dynamics, empowering investors and analysts with insights for informed decision-making.

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
