# OpenReview forum: "Analyzing Complex Interdependencies in Financial Markets: A Neural Network-Based Approach for News Impact Assessment"
_ICLR.cc/2024/Conference — Submitted to ICLR 2024_

### Official Review · Reviewer_vZ8q · 2023-10-20

**Soundness:** 1 poor
**Presentation:** 1 poor
**Contribution:** 1 poor
**Rating:** 1
**Confidence:** 5

**Summary:**

The paper introduces a neural network-based method for predicting stock trends by analyzing company interdependencies and news impact.

**Strengths:**

None. The paper significantly misses the mark for ICLR standards, displaying a conspicuous absence of the analytical depth and methodological rigor that characterizes scholarly research. Its presentation and investigative approach lack the innovation, thoroughness, and scholarly discourse expected in an academic publication.

**Weaknesses:**

W1: Lack of Specificity and Detail. Several sections of the paper could benefit from more detailed information, such as specific algorithms used, neural network architecture, and a clearer explanation of the variables included in the model. The broad statements and lack of detailed data or theoretical backing make it difficult to fully assess the validity of the claims.

W2: The paper lacks a comparison with existing models or approaches. This omission makes it challenging to gauge the actual advancement this research proposes.

W3: The paper fails to clearly convey complex procedures, particularly around the data collection and neural network modeling. This lack of clarity could hinder readers' understanding.

W4: The research's basis on a hypothetical situation, rather than real-world data and scenarios, may detract from its applicability and relevance. It creates a simulation-like environment which may not account for all real-world variables and uncertainties, decreasing the robustness of the findings.

W5: The paper lacks a thorough literature review, which is crucial for situating any research within the context of existing knowledge. Additionally, there are very few references (only two!), and those included are not from peer-reviewed sources, which could call into question the research's grounding in established academic discourse.

**Questions:**

Q1: This paper requires substantial revisions and enhancements in many aspects (presentation, methodology, experiments, discussion, etc.) to align with the stringent standards expected by top-tier AI conferences.

---

### Official Review · Reviewer_rbKL · 2023-10-30

**Soundness:** 2 fair
**Presentation:** 1 poor
**Contribution:** 2 fair
**Rating:** 1
**Confidence:** 5

**Summary:**

This submission studied a hypothetical scenario where the companies' stock prices rely not just on their progress, but also on the performance of their interdependencies (e.g., their clients, suppliers and competitors). The authors tracked stock values over time and assesses news sentiment’s influence, presented a comprehensive view of news-event-driven market dynamics.

**Strengths:**

* This submission studied an important problem, i.e., how companies' interdependencies influence their stock values.

**Weaknesses:**

* This submission is not a scientific writing, with nearly no references at all, and model designs / data source details missing. I do not think this paper should even pass the pre-screening.

**Questions:**

N/A

---

### Official Review · Reviewer_fXLD · 2023-10-31

**Soundness:** 1 poor
**Presentation:** 1 poor
**Contribution:** 1 poor
**Rating:** 1
**Confidence:** 4

**Summary:**

This work firstly collected stock news and their numerical data via stocknewsapi.com, and then applied regression analysis. Finally, it presents a picture of integrating more features into a neural network model for trend prediction. This manuscript is not ready for review because it lacks a clear research niche, a sufficient literature review, a proposed novel approach as a solution, and comprehensive evaluations to support arguments.

**Strengths:**

n/a

**Weaknesses:**

It lacks a clear research niche, a sufficient literature review, a proposed novel approach as a solution, and comprehensive evaluations to support arguments.

**Questions:**

n/a

---

### Meta-Review · Area_Chair_acav · 2023-12-08

**Metareview:**

This paper looks at the problem of stock market prediction and conducts a study by collecting data and running regression. The paper does not fit the requirements of the ICLR conference, as it does not present novel machine learning research, and nor does it contain rigorous experiments applying machine learning to solve a clearly defined research question. It also lacks citations to and comparisons with related research.

**Justification For Why Not Higher Score:**

All reviewers recommend rejection, and the authors did not respond to the reviewers' comments and questions.

**Justification For Why Not Lower Score:**

N/A

---

### Decision · Program_Chairs · 2024-01-16

Reject